# Characterization and Analysis of the Mortars of the Church of San Francisco of Quito (Ecuador)

**M. Lenin Lara Calderón** [1,2,*], **Inés del Pino** [3], **Sol López-Andrés** [4] **and David Sanz-Arauz** [1]

1   Department of Construction and Architectural Technology, Polytechnic University of Madrid, 28040 Madrid, Spain; david.sanz.arauz@upm.es
2   Faculty of Arts, Design and Architecture, UIDE International University of Ecuador, Simón Bolívar Av., Jorge Fernández Av., Quito 170411, Ecuador
3   Faculty of Arts, Design and Architecture, Pontifical Catholic University of Ecuador, Quito 170121, Ecuador; idelpinom@puce.edu.ec
4   Department of Mineralogy and Petrology, Complutense University of Madrid, 28040 Madrid, Spain; antares@ucm.es
*   Correspondence: lenin.lara.calderon@alumnos.upm.es or mlara@uide.edu.ec; Tel.: +593-994384851

**Abstract:** The relevance of the Franciscan community is reflected in the San Francisco church in Quito, which was built between 1535 and 1755. This architectural work belonging to the Franciscan complex was implanted on a plot of land with an area of 3.5 hectares and was one of the first buildings in the Audience of Quito. Eleven mortar samples that covered the walls of the central nave and side chapels were taken from the church's main temple. The procedure proposed by the authors is based on a combined methodology following the standards and protocols for the less-invasive extraction of heritage samples. Tests included X-ray diffraction, petrography, and scanning electron microscopy with a microanalysis of the samples. Mortars with a rustic composition and rough manufacturing were identified to differentiate two types of mortar, one of earthen with volcanic aggregates and another of lime with volcanic aggregates. The mining data validated the existing historical documentation, the imaginary process, and the stages of the established constructions.

**Keywords:** lime mortar; earth mortar; mortars with volcanic aggregate; mineralogy of historic mortars; Quito; cultural heritage; church of San Francisco of Quito





## 1. Introduction

Quito and its historic center have two areas of heritage protection. One, called the first-order area, is 54 hectares, and the second, called the protection area, is 376 hectares. In this area, there are 4674 built properties [1,2]. The 1888 Quito map, drawn by Gualberto Pérez, shows 17 religious buildings in the first-order area, giving it a conventual character [3]. Years later, in 1978, the city was distinguished as a World Heritage site by UNESCO (Figure 1). The arguments for this award centered around the unity of architecture and the landscape and the human diversity that was concentrated in this urban space [4].

After the Spanish city was founded, the first religious communities to arrive in Quito were Franciscans, Mercedarians, Augustinians, and Dominicans. Each group was assigned and delimited a site in the parish of El Sagrario during the sixteenth century. The Franciscan site is directly related to the cultural pre-existence of the pre-Hispanic market that occupied the first-order area of the historic center. The San Francisco square, the Plaza Mayor square, and the Santo Domingo square were apparently the vertices of the ancient market, located in a complex topography, with two deep ravines and three hills that enclosed the pre-Hispanic site [5]. In this space of symbolic pre-existence, the Spanish city was founded in 1534.

The Franciscans settled in Quito in January 1535. By mid-1538, the Franciscans were in possession of the current property, which occupies a full block and is bordered on the

north by Diego Mideros Street, on the south by Simon Bolivar Street, on the east by Cuenca Street, and on the west by Imbabura Street. The surface of the property is 3.5 hectares; in front of the building, there is a square that has the same name and an approximate surface of 0.74 hectares, including the atrium, which is a space that uses the old street.

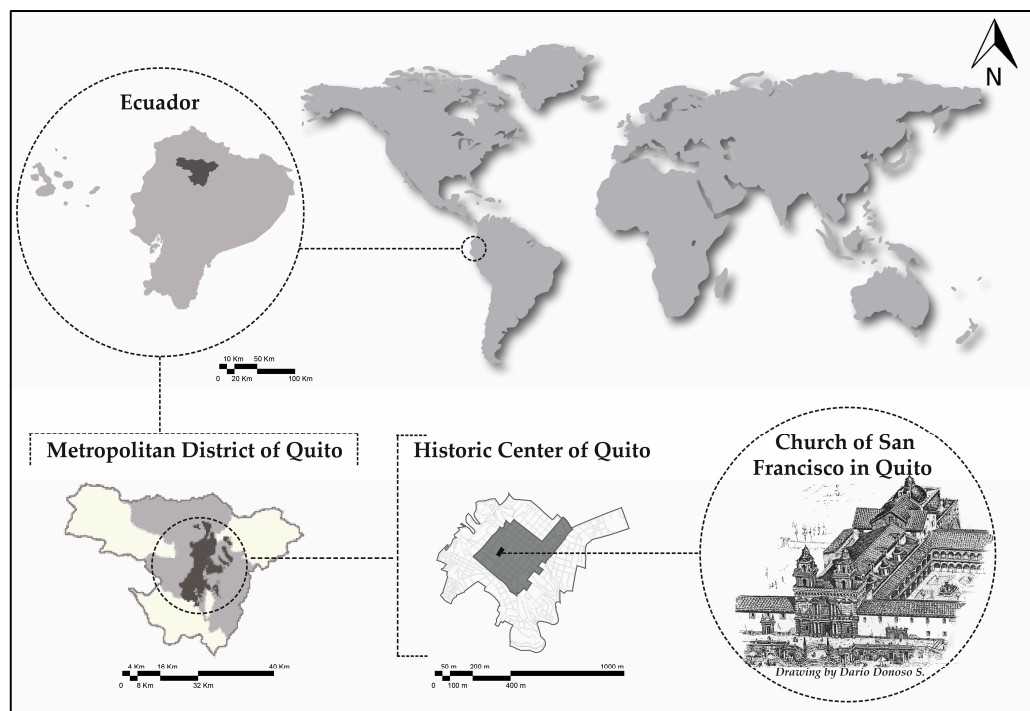

**Figure 1.** Study site supported by an illustration by Dario Donoso Samaniego, page 92 [6].

*Historical Description of the Construction of the Temple*

The first construction of the temple was modest and used local materials such as adobe, wood, and straw. The construction was overseen by the Flemish Franciscan Fray Jodoco Ricke, who made it an important point of evangelization for the indigenous people of Quito. Gento Sanz documented that in 1555, the roof fell down. Consequently, mass was no longer celebrated. In view of this loss, help was requested from Spain, where immediate aid was authorized so as not to stop the evangelization process [7].

In 1570, Fray Jodoco Ricke was appointed as Popayan to begin the works of the order, while in Quito, these works remained the purview of other religious individuals and architects until 1573 (Figure 2). The construction of the second church began in parallel with the first church, called "the big church", the first stage of which was completed in 1583. The normal development of the work was interrupted by the earthquake of 1587, which forced the reconstruction of the building due to significant damage and collapses in important areas.

Notably, two churches were built simultaneously on the lands of the order within only a few years: a small rustic church in use and a second, larger church, which, at that time, was under construction. Likewise, two cloisters, the main cloister and the museum cloister, were constructed almost simultaneously. The authors mention that local architects and master builders were involved in the construction and had to manage different activities within the complex [8].

In 1583, the consolidation of the grand church began, featuring a design comprising a central nave and side chapels. In particular, the central nave impresses with its magnificent proportions, height, and beautiful Moorish coffered ceiling, although it remains uncertain whether the church's design matches that of the transept. On both sides, the lateral naves consist of five chapels, each adorned with domes and lanterns and boasting exquisite wooden altarpieces. The church features a transept supported by four toral arches con-

structed on pilasters adorned with painted stucco lacery. The initial phase of building this grand church was undertaken by two skilled master builders: Gaspar de Borges, a talented Quito mason, and Francisco Benítez, an accomplished master builder from Portugal [9].

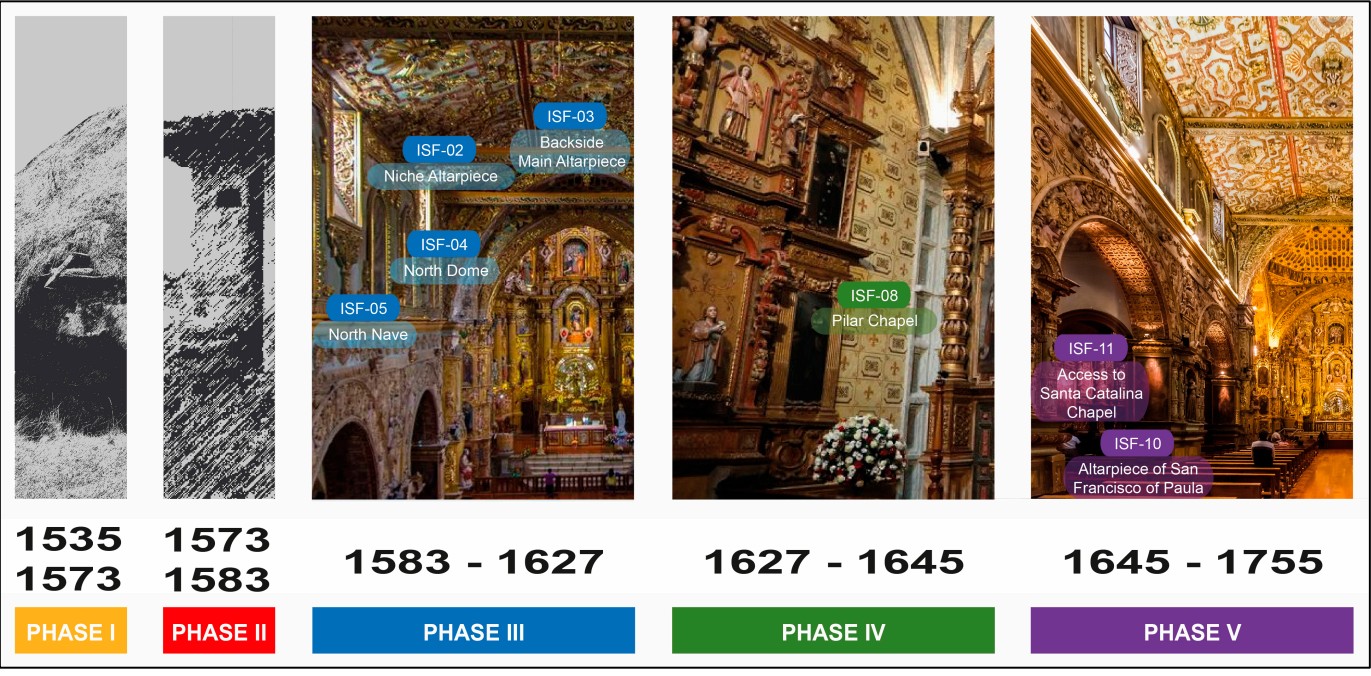

**Figure 2.** Representation of the five construction phases of the church of San Francisco over 220 years. Phase I (1535–1573) earth and straw construction; (1573–1583) phase II; (1583–1627) photographs of the south side of the central nave; (1627–1645) a photograph of the entrance to the Chapel of the Pilar; and (1645–1755) a photograph from the central nave. The image describes the location of some of the mortar samples extracted.

According to Susan Webster, between 1618 and 1627, the main entrance was enlarged to the size we know today. In 1623 and 1645, the Moorish coffered ceiling of the transept, the skylights of the side chapels, the half-orange dome of the main chapel, and the coffered ceiling of the main nave were completed [10]. During this time interval, the complex suffered two eruptions of the Pichincha volcano in 1582 and 1660, which damaged the central nave, the lateral chapels, and the towers.

Between 1627 and 1645, the spaces of the temple were strengthened in greater detail. The presbytery was finished between 1623 and 1624, and Francisco Fuentes built the half-orange dome of the main chapel between 1623 and 1625. During these years, the Moorish coffered ceiling of the central nave, transept, and ribbed ceiling were built. In 1645, the skylights of the side chapels were built by Francisco Fuentes [11].

In this period, there was litigation between the brotherhood of Santa Catalina and the heirs of Atahualpa that began in 1621 and ended in 1640, during which the brotherhoods argued to be owners of the chapel, which has a painting alluding to the saint, while the heirs noted that they ordered the painting to be made expressly for this place given their devotion of the deceased to Santa Catalina.

This quarrel was related to the increase in demand for burial sites at the church, the difficulty of the order in fulfilling all requests, and the discussion regarding the burial site of one of the heirs of the Inca Atahualpa at the time when the organization of the church changed the main entrance to the west, facing the square. The religious complex would be consolidated around 1755 when the Franciscan complex was seriously affected by an earthquake that took place in April. This event had one of the biggest impacts on Quito in the history of the city due to the number of deaths and material losses.

According to the account given by the Franciscan Gento Sanz, the church suffered extensive damage, including the loss of its roof. The suspenders of the Moorish coffered ceiling became detached from the wall supports, the chapels lost their domes and lanterns, and the main dome collapsed. Additionally, the towers lost their spires, while various cells, the cloister, and other internal spaces within the Franciscan complex became uninhabitable. Consequently, the works that had been recently finished were affected, and others collapsed. Some repairs were made to the church after the earthquake of 1755 by Fray Antonio de Jesus Bustamante, who governed the province in 1803. According to his criteria, it was necessary to protect the building, restore the damage, and repair the church promptly, giving priority to the towers, facades, and temple [12,13] and relegating additional repairs or constructions to the convent and cloisters of the block; in other words, the repair work had to wait for about 48 years.

This manuscript is part of a larger research project, which is intended to identify the chemical and mineralogical characteristics of the mortars used in the construction of the San Francisco de Quito church, of which there are no previous studies on the characterization and analysis of the temple mortars. Previous studies in 2003 of the maximum convent adjacent to the study area were carried out; however, current research will serve as a technical basis for future interventions in this building or those belonging to the viceregal period.

## 2. Materials and Methods

### 2.1. Samples of the Case Study

In this case study, 11 mortar samples were taken and analyzed throughout San Francisco Church, supported by historical information and following the guidelines of the technicians who maintain the property, as shown in Figure 3.

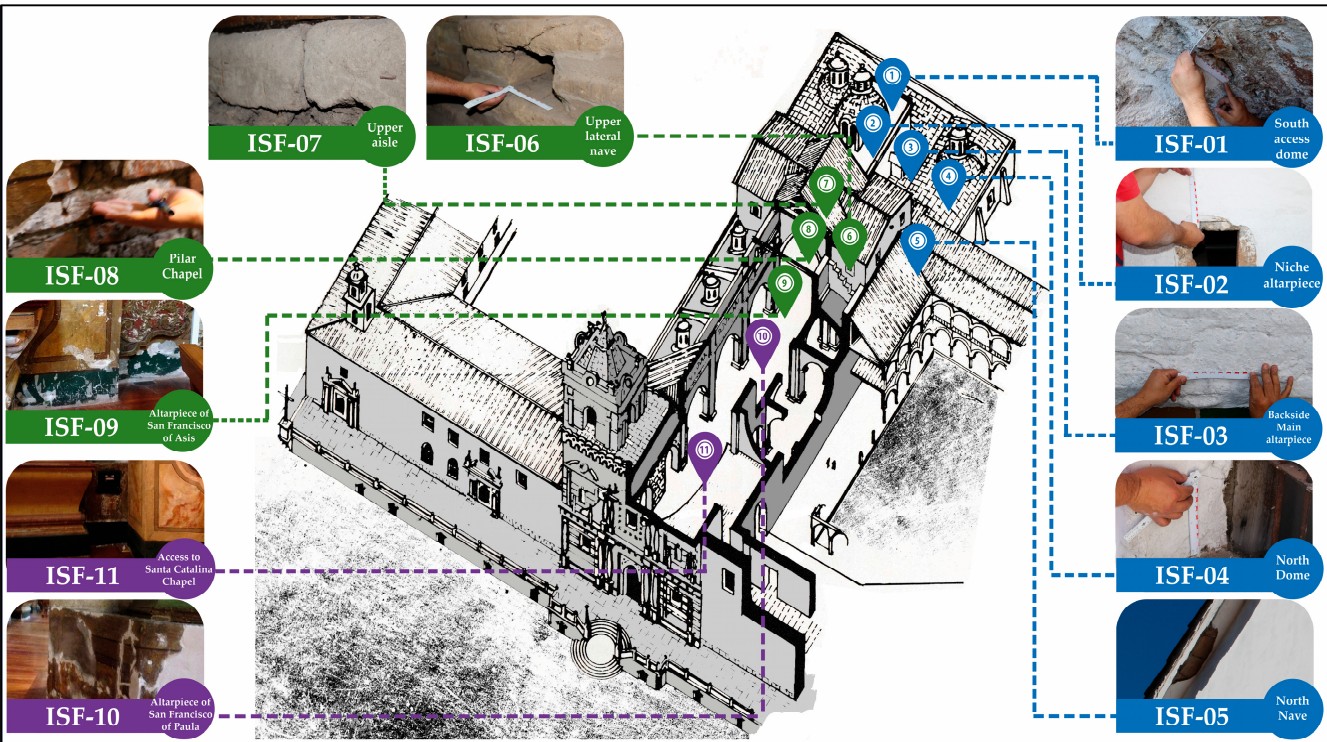

**Figure 3.** Interior axonometry of the San Francisco church with the locations of the 11 extraction points analyzed. ISF-01 South access dome; ISF-02 Niche altarpiece; ISF-03 Backside main altarpiece; ISF-04 and ISF-05 North Dome; ISF-06 Upper lateral nave; ISF-07 Upper aisle; ISF-08 Pilar Chapel; ISF-09 Altarpiece of San Francisco of Asis; ISF-10 Access to Santa Catalina Chapel; ISF-11 Altarpiece of San Francisco of Paula.

Historically, the church of San Francisco recorded a provisional construction phase as phase I (1535–1573) and a later period of construction as phase II (1573–1583), for which we do not have physical samples for analysis.

For phase III (1583–1627), the samples were ISF-01, ISF-02, ISF-03, ISF-04, and ISF-05. These samples, numbered 1, 2, 3, 4, and 5, were extracted from the upper part in different places: the first in the dome on the south side, the niche of the main altarpiece, the back wall of the main altarpiece, the side wall of the north dome, and the cover of the north side, respectively.

For phase IV (1627–1645), the samples were ISF-06, ISF-07, ISF-08, and ISF-09. Samples 6 and 7 were taken from the north lateral nave in the upper area. Sample 8 was taken from the first floor on the southwest wall in the Pilar chapel near the access to the catacombs, and sample 9 was taken on the first floor above the lower area of the San Francisco Altarpiece.

For phase V (1645–1755), the samples were ISF-10 and ISF-11. Both samples were extracted from the altarpiece in the southern lower part.

### 2.2. Methods of Identification

The present investigation adapted a combined methodological procedure that merged qualitative and quantitative variables with analytical data. Among the inputs were the historical documentation and locations of the points according to their chronological phases. With this previous knowledge of the construction history of the temple, the location of the extraction of the samples was determined. Regarding the historical aspects, variables such as the historical construction process, construction phases, practices, and traditions in construction systems were studied, as well as the main alterations and transformations suffered by the temple due to the seismic vulnerability of Viceregal Quito [14,15].

In this same historical analysis, existing historical graphic documents were emphasized, including manuals, engravings, photographs, and blueprints, which provided historical information on the construction of these elements accompanied by semi-structured interviews with experts, such as historians, technicians, and master builders, on the subject.

Analysis of the site began with visual recognition of the places related to the historical components, pointing out sectors where the mortar would be extracted. In this way, it was possible to differentiate and establish the number of mortar layers, the colors of the sample, the marks left by external elements or organic materials such as straw, and the particularities of each coating to be extracted, as well as the nature of its binder or aggregate used and the addition of materials.

Inside the building, once historical documentary information was corroborated, the type of cladding was visually identified, and the sampling points were located, for which UNE-EN 16085 was used [16]. This methodology was adapted due to the lack of a historical material catalog to compare against the samples in this study case.

In this part of the process, as identified in Figure 4, the previously justified documentary matrix was relied upon and used to simplify the historical documentation, the places of sampling, and the characteristics of the samples at the time of their registry. We also ensured that samples were free of constructive pathologies or marks that altered their readings, as such data provide clear information about the sample and are used for later refinement and tabulation [17,18]. Notably, the investigation faced samples of volcanic andesite with coarsely manufactured clasts several thickness layers in diameter in a sandy matrix of fine–medium grains [19–21].

The relevance of this research, in addition to the qualitative data resulting from the documentary matrix, relied on the semi-quantitative data provided by the archaeometric analysis. After identifying and cataloging each of the samples, the decision was made to analyze the historical mortar samples in a certified research center due to the limitations of local laboratories with regard to the resulting sample size, processing time, cost of analysis, and comparative interpretation of the results.

For identification and characterization of the samples, several tests were initiated. Thus, small fragments of all samples were examined for initial analysis with a binocular

stereo microscope Zeiss Stemi 305, and thin slices of some samples (from phase III, sample ISF-03; from phase IV, samples ISF-06 and ISF-08; and sample ISF-11 from phase V) were visualized using a Zeiss Primotech petrographic microscope. A homogeneous and representative fraction of 2 g of sample was ground and sieved at 53 μm [23–25] for the identification of the mineralogy by X-ray diffraction (XRD) in a Bruker D8 ADVANCE diffractometer, which allowed us to identify the relative proportions of each mineralogical phase. For this purpose, we used the Chung method [26], while data analysis was supported by Bruker EVA software, which retained experimental errors of plus or minus 5%.

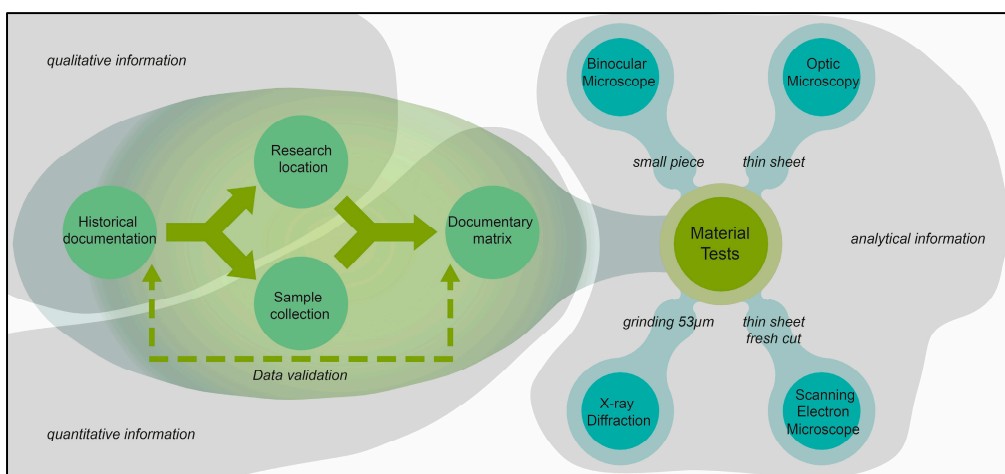

**Figure 4.** Graphical synopsis of some of the methods used in the research process of this study [22].

Thin slices of selected samples (ISF-03, ISF-06, ISF-08, and ISF-11) were used for the morphological and chemical analysis by scanning electron microscopy (SEM-EDX) [27–29].

Notably, in this research, characterization of the historical mortars was carried out using chemical–mineralogical analysis. Due to the church's status as a living museum of religious art and patrimonial property, it was not possible to extract samples of the coating in a greater volume for mechanical and physical tests. Due to these limitations, such samples were not considered within the scope of this research.

## 3. Results

In San Francisco, a historical comparative analysis was carried out in 2003 by Antonia Moropoulou on the Convent Maximus of the religious order adjacent to the church on its north side, which aimed to study the brick and stone walls and some joint mortars [30]. For this study, the focus was on the main temple, which was differentiated by the chronological phases described in Figure 2. After crystallographic identification, these phases are visualized in Table 1. The mineral phases of the samples were identified via XRD.

POM and XDR carried out on the samples allowed us to identify the mineralogy used in the manufacture of the mortars in the temple. Table 2 describes the classification of the mortars found in the three recognized construction phases that comprise the studied chronology.

This first reading of the results and identification of the earth and lime mortars with volcanic compositional characteristics demonstrate the geological nature of the Andean zone of Viceregal Quito, as well as the adaptation of the construction on the site since the colonizers did not have all the materials to place their works, assuming that the local material was used consistently by the site workers and the spatial manufacturing was that to which the colonizers were accustomed.

Figure 5 shows an illustrative example of the unidirectional patterns of X-ray diffraction that allowed the several intensities of each mortar to be visualized. Thus, the lime mortar with volcanic aggregates can be found in samples ISF-03 and ISF-11, while the earth mortar with volcanic aggregates can be visualized in samples ISF-06 and ISF-08 based on the analyses carried out in the main temple of the San Francisco church.

**Table 1.** Mineralogical identification and quantification in % with XRD. The symbols used for the mineralogical phases are presented according to [31]. Quartz = Qz; plagioclase = Pl; amphibole group = Amp; calcite = Cal; illite group (clay minerals) = Ilt.

| Samples | Qz | Pl | Amp | Cal | Ilt |
|---------|------|------|------|------|------|
| ISF-01 | 2.0 | 62.0 | 11.0 | 18.0 | 7.0 |
| ISF-02 | 3.0 | 81.0 | 5.0 | 7.0 | 4.0 |
| ISF-03 | -- | 70.0 | 13.0 | 14.0 | 3.0 |
| ISF-04 | 1.0 | 76.0 | 11.0 | 10.0 | 2.0 |
| ISF-05 | 2.0 | 77.0 | 13.0 | 6.5 | 1.5 |
| ISF-06 | 6.5 | 71.5 | 19.0 | -- | 3.0 |
| ISF-07 | 2.0 | 84.0 | 7.0 | -- | 7.0 |
| ISF-08 | 2.0 | 89.0 | 6.0 | -- | 3.0 |
| ISF-09 | 3.0 | 73.0 | 19.0 | -- | 5,0 |
| ISF-10 | 2.0 | 67.0 | 16.0 | 12.0 | 3.0 |
| ISF-11 | 2.0 | 75.0 | 3.5 | 13.0 | 6.5 |

Due to existing limitations in this investigation, it was not possible to perform a clay analysis on the samples ISF-06, ISF-07, ISF-08, and ISF-09.

**Table 2.** Description of the ancient mortar samples of San Francisco.

| Construction Phase | Samples | Mortar Type | Location |
|---------|---------|-------------|----------|
| PHASE III | ISF-01 | Lime | South access point of the dome |
| | ISF-02 | Lime | Niche altarpiece |
| | ISF-03 | Lime | Backside main altarpiece |
| | ISF-04 | Lime | North dome |
| | ISF-05 | Lime | North nave |
| PHASE IV | ISF-06 | Earth | Upper lateral nave |
| | ISF-07 | Earth | Upper aisle |
| | ISF-08 | Earth | Pilar Chapel |
| | ISF-09 | Earth | Altarpiece of San Francisco of Asis |
| PHASE V | ISF-10 | Lime | Access to Santa Catalina Chapel |
| | ISF-11 | Lime | Altarpiece of San Francisco of Paula |

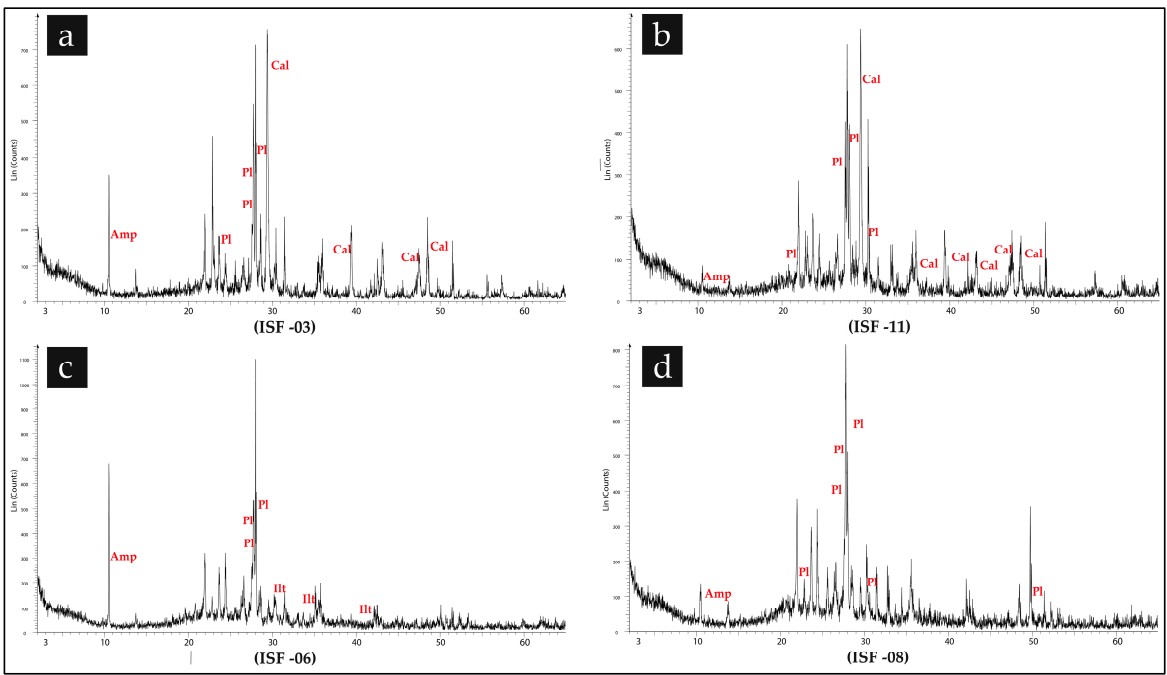

**Figure 5.** XRD diffraction patterns: (**a**,**b**) samples ISF-03 and ISF-11 lime mortars with volcanic aggregates; (**c**,**d**) samples ISF-06 and ISF-08 earth mortars with volcanic aggregates.

Optical and electron microscopy observations on thin sections of the representative samples ISF-03 from phase III and ISF-11 from phase V (shown in Figures 6 and 7, respectively) allowed us to identify a high percentage of plagioclase and amphibole and quartz in a lower percentage, as well as a conglomerate matrix formed by small calcite grains and clay minerals. These observations allow the description of the rustic characteristic of the volcanic composition from which the materials originate [32,33].

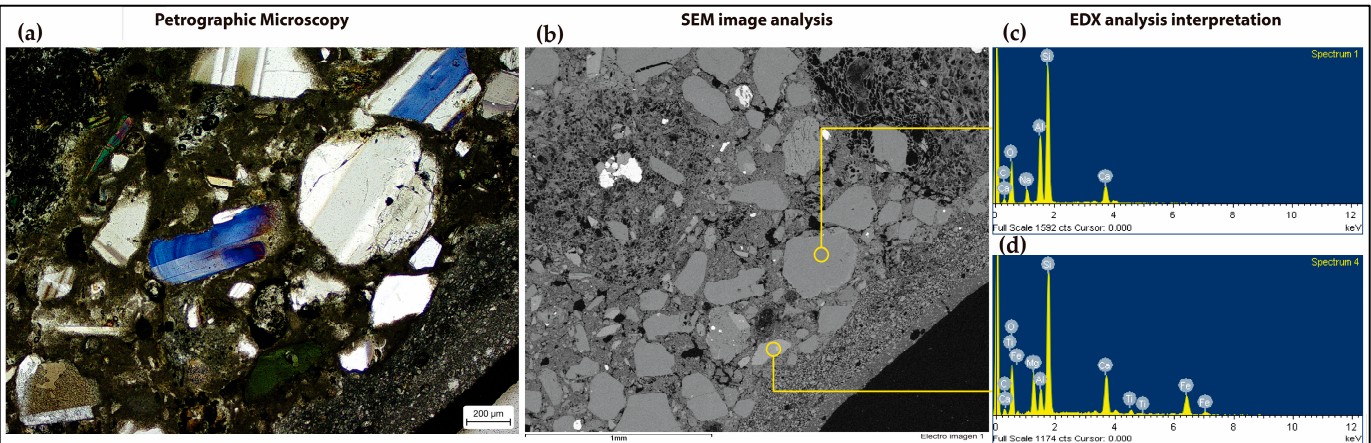

**Figure 6.** Thin section of sample ISF-03: (**a**) POM (with crossed nichols) image at 50× magnification, which allows one to visualize the volcanic aggregate with grains of plagioclase with polysynthetic twins and grains of green amphibole and quartz; (**b**) SEM image with BSE at 20× magnification; (**c,d**) EDX analysis supported by a BSE image of lime mortar with a volcanic aggregate of a plagioclase and an amphibole, respectively.

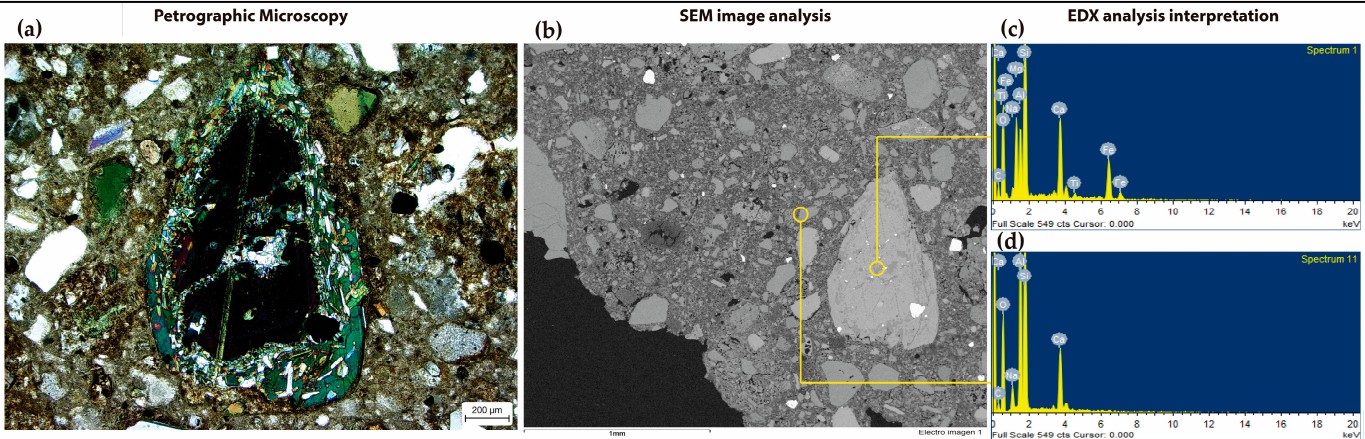

**Figure 7.** Thin section of sample ISF-11: (**a**) POM (with crossed nichols) image at 50× magnification, with a large amphibole crystal zoned; (**b**) SEM image with BSE at 20× magnification; (**c,d**) EDX analysis supported by a BSE image of lime mortar with a volcanic aggregate of an amphibole and a plagioclase, respectively.

Figures 8 and 9 show the morphological and compositional observations by SEM of samples ISF-06 and ISF-08 of phase IV of the chronological hypothesis. These samples present a volcanic aggregate but without the lime binder used in the other construction phases, resulting in an earth mortar with volcanic aggregate [34].

The optical observation of the sample ISF-03 (Figure 6) showed that it is a mortar whose conglomerate matrix is calcite, where aggregates are fragments crushed from an ancient volcanic rock, probably andesite, composed of plagioclase (large yellow grains) and amphiboles (more geometric greenish in color). The EDX analysis of this sample comprised three sites of interest, reaching a total of 36 analyses that visualized several chemical elements. For this case, two relevant minerals (amphibole and plagioclase) were

identified: (b) spectrum 1 with Na, Al, Si, Ti, Mg, Fe, and Ca, and (c) spectrum 11 with Si, Al, Ca, and Na.

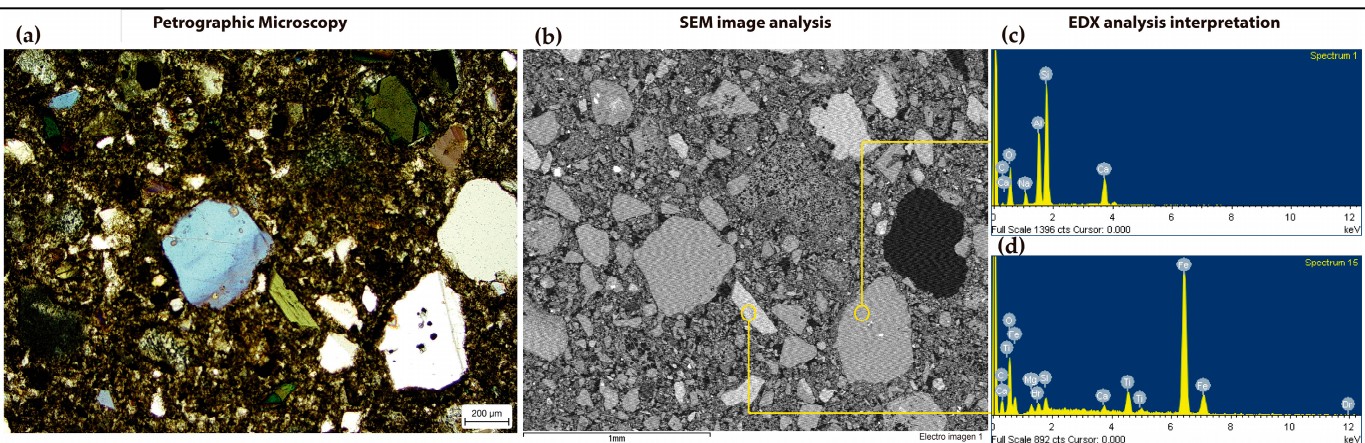

**Figure 8.** Thin section of sample ISF-06: (**a**) POM (with crossed nichols) image at 50× magnification; (**b**) SEM image with BSE at 20× magnification; and (**c**,**d**) EDX analysis of earth mortar with a volcanic aggregate of a plagioclase and an amphibole, respectively. In this case, optical observation of the sample ISF-06 (Figure 8) with a 50× magnification showed the properties of an earth mortar of phase IV that was collected from the southwest wall of the Pillar Chapel, from the access to the catacombs. Forty-seven chemical point analyses were performed on this sample by EDX. Two of them are shown, corresponding to a plagioclase (**a**) spectrum 1 with Si, Al, Na, and Ca and a small amphibole crystal (**b**) spectrum 15 with Si, Fe, Ti, Mg, and Ca.

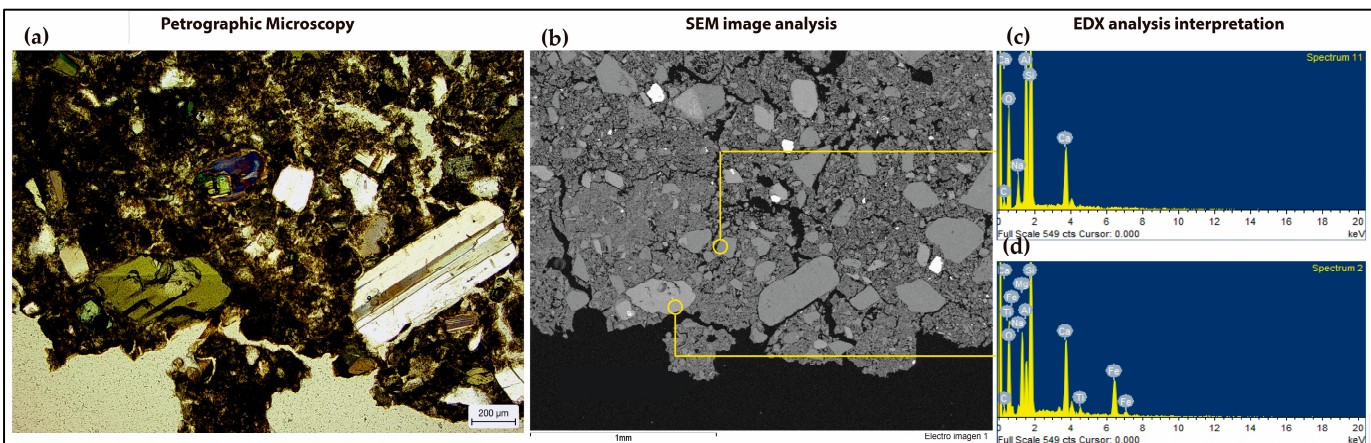

**Figure 9.** Thin section of sample ISF-08: (**a**) POM (with crossed nichols) thin section image at 50× showing an earth mortar with a fine-grained volcanic aggregate plagioclase and amphibole; (**b**) SEM image with BSE at 20× magnification; and (**c**,**d**) EDX analysis of a plagioclase and an amphibole, respectively. Finally, we have another example of earth mortar with a volcanic aggregate, sample ISF-08 (Figure 9). This sample was extracted from the southwest wall of the Pillar Chapel from the access to the catacombs; green fragments of amphibole and twinned crystals of plagioclase were observed; 68 EDX analyses were performed. Two representative grains of the total plagioclase and amphibole composition have been chosen: (**a**) spectrum 11 with Al, Si, Na, and Ca, and (**b**) spectrum 2 with Si, Mg, Fe, Al, Ti, Na, and Ca.

Another sample of lime mortar with volcanic aggregate is ISF-11 (Figure 7), which retained the same observation parameters under 20× magnification. This sample was extracted from the south wall access to St. Catherine's Chapel. We visualized, as in the previous case, a calcite-based mortar with fragmented aggregates of andesite, observing plagioclase and amphibole. EDX analysis on this sample was carried out at several points

of interest, reaching a total of 73 analyses. For this case, spectra 1 (a) corresponds to an amphibole with Na, Mg, Al, Si, Ca, Ti, and Fe, and spectrum 11 (b) corresponds to a plagioclase with Al, Si, Na, and Ca.

## 4. Discussion and Interpretations

Considering that this church is one of the oldest temples built in the viceroyalty of Quito, in previous publications on Santo Domingo's church, the volcanic nature of the aggregates in the area was identified, while the Company of Jesus' church, one of the later churches built at that time, highlights the discipline of the order that allowed it to experiment with finer and more delicate mortars, such as those made of lime and plaster, as corroborated by previous research [17,35,36].

The identification and characterization of the samples analyzed in the San Francisco church allow us to conclude the volcanic origins of the stone aggregate of the mortars. In this way, the lime mortar aggregates show the vitreous texture and rustic manufacturing of the masonry, joint mortar, and earth lining [37], the latter with the absence of calcite in its composition. In this masonry, the conglomerate is constituted by minerals of the clay group and traces of quartz and amorphous phases, validating some of the research conducted in the equatorial Andean region [38,39].

Interpretation of the results determined that the 11 samples had minimum proportions of quartz that fluctuated between 1% and 6.5%. This mineral was not detected in the ISF-03 sample. For calcite, ranges between 6.5% and 18% were observed, with the exception of the samples ISF-06, ISF-07, ISF-08, and ISF-09 of earth mortars that did not identify calcite [40]. The quantities of plagioclases, amphibole, and minerals of the clay group have logical values with respect to the nature and type of mortars.

According to historical documents, the extraction quarry site was the farm of the nuns of the Santa Clara Convent in the Pintag Tolontag sector, designated in 1551 to Francisco Ruiz by the power of the viceroyalty to the Franciscan order. This Andean moor has several centers of andesitic composition that are in the range of 56–61% $SiO_2$ and 1–2% $K_2O$ [21,22,30].

In the San Francisco church, an analysis of optical microscopy was carried out on the chosen thin sections. The samples were as follows: ISF-3, ISF-6, ISF-8, and ISF-11. The results allowed us to observe the different morphologies, textures, sizes, and shapes of the aggregates present. The SEM analysis identified the presence of a calcic-sodic plagioclase and an amphibole with Ca, Na, Al, Fe, Mg, and Ti as fundamental aggregates and clay minerals and calcite as binders.

Two types of mortar were identified. One was lime with coarse-grained volcanic aggregates, which is very commonly used on surfaces that feature mural paintings, narratives, and images of the religious order, such as the apse, transept, and upper linings of the central nave, as well as the lower areas of the stem wall. The other was earth mortar with volcanic aggregates, which is very common in areas with little access, where the surface does not present greater detail or finish and was evidenced in the high walls between the coffered ceiling and abutments, where the pairs of the framework were assembled and supported. This finish reflects the same coarse, rustic composition as the constructions of the Andean area that were already identified in previous investigations [30,34].

This constructive particularity of not having a better surface finish with lime and leaving the earthen wall exposed is due to the economic resources that the religious orders had access to during the construction process in America [41]. Moreover, the Franciscan community in Viceregal Quito built the entire religious complex at one time without prioritizing the construction of certain spaces according to need [8,42]. This particularity allows one to distinguish between diverse common typologies in the construction of the complex and hypothetically determine the period of the temple, all substantiated by the manufacturing process of the volcanic aggregate.

In the samples of the San Francisco church, temporality and the origin of the material applied are strongly reflected according to the construction stage and the economic realities

of the order. In addition, more than lime was used as the final coating solution applied to the surface, so this mixture offered greater durability, workability, and impermeability in addition to improving the formal appearance of the structure [43]. Both religious orders and people had economic limitations, which forced them to experiment with a rustic single-layer mortar based on the earth [19].

## 5. Conclusions

In the implantation of the Franciscan order, the locus of power after the conquest is evidenced by the replacement of this pre-Hispanic space rich in local imaginary tianguez ("central place" to trade articles among the aborigines) [44] with a new space of monotheistic worship. Inside the church, there is evidence of the material and spatial adaptability of locals and foreigners to the new requirements of the viceroyalty.

Multiple assumptions historically transmitted by chroniclers and historians on the construction of this space by workers were validated, including the reciprocal transference of ancestral knowledge through sampling analysis, checking the volcanic rustic origin of the materials used, and the adaptation to the construction and aesthetic requirements of the temple and religious order applicable to each determined construction phase.

The sampling analysis allowed us to assess the origin of the aggregates and confirm that the mortars were supplemented with local raw material and volcanic–calcitic aggregates, as corroborated by the results of the analysis of the mineralogy by tXRD, POM, and SEM.

Based on the findings (comparing historical data and new petrographic research from this area), it is quite possible to corroborate the hypothesis that the Pintag quarry was the deposit that provided material for the temple. Although the material from the original quarries is currently not accessible, historically, it is known which quarries were, but they are currently closed, and erosion has eliminated them. However, as the research progresses, an attempt could be made to search for original material to study its evolution over time and confirm the origin of the mortars used in this case study.

Characterization of the samples by XRD and SEM-EDX showed similar compositional patterns, with plagioclase as the most abundant mineralogical phase. Amphibole, clay minerals, calcite, and quartz are minerals that are found in smaller proportions in the lime mortars of constructive phases III and V. Phase IV presents mineralogy compatible with earth mortars due to the absence of calcite. Thanks to these analyses, several tales of chroniclers and previous studies that are part of this research were confirmed.

The mineralogical nature of the adobe masonry samples ISF-06, ISF-07, ISF-08, and ISF-09 provide information on the quality of the mud material, as these materials are still present today, despite the high vulnerability of adobe and its mortars.

Future research resistance tests may be carried out on replacement mortar samples made using the original material from the quarry or a similar source, as well as stone samples from the facades of the Viceregal Quito temples.

Finally, beyond the composition of the San Francisco church mortars, their volcanic nature in all the construction phases identified in this research reflects the adaptability of the material and the techniques applied for the monumental construction of the temple, which withstood strong telluric movements in Colonial Quito in 1755, 1797, 1859, and 1868. Moreover, the reforms carried out in the past and in the present respect the particularities of these mortars.

**Author Contributions:** Structure, M.L.L.C.; methodology, M.L.L.C., D.S.-A. and S.L.-A.; analysis, M.L.L.C., D.S.-A. and S.L.-A.; validation, D.S.-A. and S.L.-A.; historical analysis, M.L.L.C. and I.d.P.; research, M.L.L.C.; resources, M.L.L.C.; data curation, M.L.L.C., D.S.-A., S.L.-A. and I.d.P.; writing—preparation of the original draft, M.L.L.C., D.S.-A., S.L.-A. and I.d.P.; writing, revision, and editing, M.L.L.C., D.S.-A., S.L.-A. and I.d.P.; translation, M.L.L.C.; visualization, M.L.L.C.; supervision, D.S.-A. and S.L.-A.; project administration, M.L.L.C.; funding acquisition, M.L.L.C. All authors have read and agreed to the published version of the manuscript.

**Funding:** The dissemination of the results of this research stage was financed by the International University of Ecuador project UIDE-DGIP-MAT-PROY-22-005. This article is part of research in the DTCA Doctorate Program in Construction and Architectural Technology of the Polytechnic University of Madrid, whose subject is "Characterization of the mortar coating of Colonial Quito in the 16th, 17th and 18th centuries".

**Data Availability Statement:** The historical books in this document are primary sources that have not been updated since the middle of the 20th century. Some are considered 'gray literature' but serve as a fundamental basis for the qualitative investigation of the present manuscript.

**Acknowledgments:** We thank the openness and ease of access and sampling of the church in 2018 and its authority as the church of San Francisco, especially Arch. Héctor Vega Quinteros, Friar Jaime Zhindón Minchala, and Luis Bastidas, Guard of the Franciscan temple. This analysis was carried out by the Geological Techniques Unit of the Support Center for Research in Earth Sciences and Archaeometry of the Complutense University of Madrid (Spain).

**Conflicts of Interest:** The authors declare that they have no conflict of interest. Sponsors had no role in the design of the study; in the collection, analysis, or interpretation of data; in the writing of the manuscript; or in the decision to publish the results.

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
