# Peer review of "Characterization and Analysis of the Mortars of the Church of San Francisco of Quito (Ecuador)"

_heritage, doi:10.3390/heritage6120393_

Round 1

Reviewer 1 Report

Comments and Suggestions for Authors

Point 1.

In the introduction the novelty of the paper should be clearly reported. At least the knowledge process, as unvaluable tool for studying cultural heritage, should be mentioned. Among the others, Author should refer to the following studies, where this topic is fully investigated:

·      Luchin, G., Ramos, L. F., D’Amato, M. (2020). Sonic tomography for masonry walls characterization. International Journal of Architectural Heritage, 14(4), 589-604.

·      Lombillo, I., C. Thomas, L. Villegas, J. P. Fernández-Álvarez, and J. Norambuena-Contreras. 2013. Mechanical charac- terization of rubble stone masonry walls using non and minor destructive tests, 266–77. Construction and Building Materials. Amsterdam, NL: Elsevier.

Point 2.

Introduction should be revised. Some parts me be condensed since information reported are not strictly pertaining to the research work.

Point 3.

In the current version the paper just explains the results of some X-rays graphs. More comments should be reported, for instance, considering other similar results already published in literature.

Point 4.

How the results published in this work may be coupled with other non-destructive tests in order to properly identifying the mortar present in the masonry analyzed? This is an important  

Point 5.

Conclusion section should be revised reporting only the main outcomes of the work presented.

Comments on the Quality of English Language

Minor editing of English language required

Author Response

Dear reviewer

The reviewer is congratulated for the recommendations suggested to improve the comprehension and reading of the research.

Thank you very much for the positive comment on the research; Latin America and especially Ecuador do not have examples of research on mortars and characterization of materials in their buildings with heritage value.

Comments that were kindly made at the same time have been revised and corrected, the document as a whole went through a revision and editing of English for service the MDPI.

Reviewer 2 Report

Comments and Suggestions for Authors

The paper titled "Characterization and analysis of the mortars of the church of 2 San Francisco of Quito (Ecuador)", examine samples of mortars from various construction phases of San Francisco Church in Quito, Ecuador, using different techniques such as MOP, DRX and SEM-EDX.

General comments:

Introduction:

The introduction does not clearly address the study's problem. It lacks a clear exposition of the problem to research, neglecting to introduce the reader to the specific problem that needs solving. Moreover, it does not provide information on any previous restoration that had been done on the temple, or if similar interventions have been carried out on other similar pre-Hispanic monuments in Quito. There is no background about similar studies developed in the region. Lastly, it is also missing a definition of the objective that this research work aims to achieve.

Methodology

Although the methodology used in this research is appropriate for this work, an English native must carefully revise the text since it is hard to understand the procedures, especially the first part, before line 202.  Perhaps, this text about the sampling in mortars (from L156 to L 201) can fit better in the Introduction (previously adapting the literature), because it is not a specific methodology used in this case, but general concepts about sampling and analysing mortars in historic monuments.

The methodology must describe with enough accuracy the methods and techniques used, and the procedures that the authors had been follow to achieve their results.

Results.

"The presentation of results in the paper is not sufficiently clear. In certain instances, the authors blend their results with comments that should ideally be reserved for the discussion section. Results should solely focus on describing the findings without referencing other authors' work from previous research. This mixing makes it challenging to distinguish between the authors' findings and those from prior studies.

Furthermore, the petrographic descriptions need enhancement, requiring more detailed descriptions of the mineralogy and sample textures. Similarly, the SEM images lack adequate contextualization.

Regarding figures, it's advisable to place them in proximity to the paragraph where they are first cited.

Discussion and interpretation.

There is no real discussion of the results since the references are not properly indicated. The authors differenced two different mortars, but I cannot easily find which differences are between them. What are the chemical or mineralogical differences between them? What samples belong to each group?

Are those results similar to other examples of similar monuments in the area? Do they use the same mortars? Which are the differences?

When results are described, it is not clear the frontier between their own results and other authors’ statements.

Conclusions

In the conclusion section, there is an assertion that is not substantiated by the data presented in this research. The authors state that the mortar materials originate from the quarries of Pintag, but this claim lacks empirical evidence within the discussion or results section to support it. The only basis for this assertion appears to be a single bibliographic reference. To strengthen this claim, it is essential to compare the findings from this study with the data provided by Moropoulou et al. (2003) in the discussion section to establish a more solid foundation for the assertion regarding the origin of the mortar materials. Is the Pintag quarry the unique one that extracted volcanic rocks in this period?

Additionally, the other conclusions, particularly those related to the classification of the mortar, should be substantiated more effectively using the results and the discussion. It is essential to provide a more robust and explicit connection between the findings presented in the study and the conclusions drawn regarding mortar classification.

References:

"Please homogenize the references according to the authors' guidelines. Use either the full name of the journal or its abbreviated form consistently throughout the paper, but refrain from mixing the two. It's crucial to adhere to the authors' guidelines for referencing and citations."

Additionally, I have some minor comments or details to address before publishing:

Abstract: please, use the comma…

L4 According to the information I recibed from the editor, there are four authors, but in the manuscript, there is only one.

Abstract:

L11. This sentence is too long. Try to use commas to divide statements.

L13-14 "Its main temple” is repeated in the same sentence

L17: in the developed..  (can be removed). Try to rephrase this sentence for better understanding. It is too long. Try to separate the different ideas. One for the mortars and one for the mineralogy. Avoid the use of personal forms such as ”we identified..” or pronoun “us”. In scientific writing, the most important things are the results not who made these results.

19: arid: in English, arid is a land that is so dry that very few plants can grow on it, not “árido”. A dessert is arid. Perhaps you want to refer to “aggregates”

After that, perhaps, the historical description of the construction of the temple can fit as an independent section, not the introduction.

L68. Rewrite this sentence to clarify. It is hard to understand

L86 according to Webster [10] – Add reference. Check the Norms for Authors for this type of cite.

L89 If we talk…. Avoid personal forms. In my opinion, they are not appropriate in scientific style (check the whole manuscript).

L93  Figure 2. I think earth is a general term without a clear scientific meaning. In this context, perhaps you refer to mud or clay. An explanation in the introduction about the meaning of the term “earth” in this context could be provided. Which are the sources of the draws? In my opinion, they can be removed since they do not have enough quality to appreciate anything…  Check the use of the Caps in the caption: Phase I (1535-1573): earth and straw; phase II (1573-1583); phase III (1583-1627): photography….

L98 to 102 rewrite or shorten this sentence. It is not clear…

L136-137. To take the samples… this sentence can be removed.

L183 I do not understand this text: “exemplified from the doctoral dissertation [17], since reference [17] it is not clear whether it is a Doctoral dissertation or a book or what, because a doctoral dissertation is made by one author only.

I believe that Figure 4 could be omitted from the paper as it does not contribute significantly to the text. This figure represents a common scientific procedure in heritage and archaeological sciences. If it is retained, please ensure that it is placed in the text after being cited, rather than before.

Regarding Line 223, I suggest considering the possibility of conducting resistance and strength tests on mortar samples made from the original quarry material or a similar source in the future. This proposal for future research could be included in the conclusion section as a valuable suggestion."L229 San Francisco church

Line 229 to 232: These are the typical statements that have to be in the introductions section.

Line 232 to 236 This sentence is too long and hard to understand.

L238 X-ray diffraction: use XRD.

Table 1. If you determined the semiquantitative % of the different phases detected by XRD, why do you not express the result with data in %? Even more, when you use this % in the discussion section. 

L239 Caption: Why did you specify the abbreviation of the mineral species since you do not use them? You can remove this and the reference. In addition, you can differentiate the type of plagioclase but you did not do it with the amphibole type. Not even XRD? Why? In any case, if you detect different amphiboles, you can specify them in the table or the results’ comments.

L 241. This paragraph fits better in the discussion since you are not commenting anything about your results and the mineral species identified, but you are comparing your results with previous research.

L247 Table 2 is not cited in the text. This table must to be changed to the Samples sub-section.

Table 2 Description of the ancient mortar samples of San Francisco church.

What is the meaning of “Consider the volcanic origin of the quarries near Quito”? I do not understand this.  

Figure 5. Increase the red labels. They are unreadable. The grey band of the “Earth mortars” is not necessary and cover partially the figure. It is very clear the difference between samples.

X-ray diffraction: use the abbreviation XRD, as you previously referred. Or not, but not mix them. Check the whole document.

L266 POM: You can define the acronym the first time you use it. In the methodology section, you refer to PM. In my opinion, you have to use POM (polarized optic microscope). Change one or another, but homogenize it.

L267 is shown in Figure 6.

Use Fig. 6 or Figure 6.  You refer to figures as Fig. or Figure in the text. Homogenize it according to the Norms for Authors, but do not mix it.

L267  allowed.. Where is the subject of this sentence? Please, revise carefully the place of the punctuation marks in the whole document. A professional English review is strongly needed.

L269 How do you can differentiate the Illite group minerals in the microscope? And why are you not detected by XRD?

L271. I do not understand what you exactly refer to with these references: [34,35], since you are describing your findings and these references are about Italian mortars. If you are referring that your mortars are similar to those Italian mortars, you can comment on it in the discussion, not in the results.

L273. Reference [36]. The same as previous comments. The lack of lime mortar is your discovery, and was it previously discovered in the work you cite?

L275 Figure 6. I suppose that the grey scale figure is a SEM figure, but is not referred to in the caption. Is taken from an area of the figure taken with microscope analysis? If this is the case, mark it the area with a circle. If not, separate both figures in a) and b) and explain them properly in the caption and the text.  The same for Fig. 7, 8 and 9.

L278 in a 200x petrographic microscope. Sure? I think this magnification is too high for these images. Is the total magnification of the microscope or does the image have this magnification? Check this value. In any case, this information is not relevant since the Fig has the scale.

L279 I do not know what you refer to with this statement: “double crystallographic variation of the sample”. Do you refer to the characteristic twins of plagioclases? In any case, improve the petrographic description of the samples.

Fig 6 to 9: You have to describe a little bit more the findings from the petrographic study of samples and from the SEM images, with textures, grain shapes, etc.

Describe why you select the different interest sites for EDX analysis, describe their characteristics and why the EDX spectrums you select are the most interesting.

Perhaps a new table could be added with the EDX analysis findings, indicating which elements are present in the different samples and in how many EDX analyses they were detected. This can allow us to know the most abundant elements and the possible presence of trace elements.   

L318 [37] and [38,39] I do not know why this must have citations, since these are your findings. In case the authors that you cited exposed similar conclusions, or findings, please, clarify it. You indicate the presence of clay minerals such as kaolinite group minerals, but in the results section, I can not find any reference to this group of minerals in the text.

L327. I miss a reference or a comment with the logical values of Plg, Amp and clay minerals for these types of mortars.

L336 add . to separate sentences…

L337 allowed us…. Allowed us is repeated three times in the same statement. 

L339 use subscript for CaAl2Si2O8

L363  church (without caps or with caps, but homogenize the text)

L363 to 369 there is no period in the paragraph. It is impossible to understand.

L 387 What is the meaning of BSM? Please, use the same abbreviations in the methodology and the document. OM? POM? Be coherent with the use of abbreviations. 

References:

Reference 24. Check this reference: “procedings of the procedings of” sound strange.

Reference 25 to 30 Check the name of the journal. They are in the abbreviated form.

Comments on the Quality of English Language

Regarding the language, English is not my first language, so I may not be proficient enough to evaluate the style and grammar thoroughly. However, I find that a perusal review by a professional native speaker or by the Language Service of the University is strongly advisable. I have identified many grammar and vocabulary mistakes. The sentences are overly long, leading to confusion, and the punctuation marks are either misplaced or missing. All of these make the text unreadable or so hard to understand what the author’s is referring to. These concerns must be addressed before publication.

Author Response

Dear reviewer

The paper has been subjected to your recommendations to improve the comprehension, readability, and impact of the research, with the parameters of the Heritage journal.

The observations that are kindly made have been considered, taking into account the limitation and scope of the research, at the same time we will have to comment that in Latin America and especially Ecuador do not have examples of research on mortars and characterization of materials in their buildings with heritage value.

Comments that were kindly made at the same time have been revised and corrected, the document as a whole went through a revision and editing of English for service the MDPI.

Reviewer 3 Report

Comments and Suggestions for Authors

-line 136: you should report the function of the mortar samples (i.e. bedding mortar, plaster etc.);

-line 239: concerning the samples ISF 6, 7, 8, 9 you should also perform the  mineralogical analysis of the kind of clay minerals because these data are important in order to understand possible causes of decay;

-line 269: concerning the optical observations in thin section, in the case of the lime mortars you should evaluate the binder/aggregate ratio and in the case of the earthen mortars the matrix/framework ratio. Moreover, concerning the aggregate you should describe the shape of the grains (roundness) in order to understand the possible provenance (from river sediments or from grinding). You speak about this information in the Discussion paragraph, but you have to report first in the Result paragraph;

Author Response

(The authors gave the same response as above.)

Reviewer 4 Report

Comments and Suggestions for Authors

The article is quite interesting and fits the type of journal and content in which it is published. However, it should be improved in some aspects for its publication. First of all the authors should make clear the place where the samples are taken, is it inside or outside the church of the monastery. In Figure 3, where the place where the samples were taken is shown, the way of marking them with a kind of drop of water is not the most appropriate.  On the other hand, many of the images provided are difficult to read either because of their dimensions or because of the type of typography used. For example, in figure 5, in the peaks of the diffractogram, the different components of the lime mortar and the earth mortar are illegible. This is also seen in the EDX readings of Figures 6, 7, 8 and 9. We would advise that these images be treated differently, e.g. grouped in such a way that the EDX diagrams can be read. The quality of figures 2, 3 and 4 should also be improved as there are parts of them that are unreadable.

A more important aspect that misleads the reader of the text is the lack of correspondence between the image and the written content. For example, in Figures 7, 8 and 9 the EDX spectrum a) and b) is cited when in the figures it is b) and c).

Finally, a couple of questions that from my point of view would help to improve the article and to reaffirm some of the explanations provided by the authors. Firstly, it is all very well to explain that materials from the environment are used and that some of them are poor, but we have to contextualize the building under study. I explain, it is a building of the Franciscan order and as many should know, this order is characterized by the vow of poverty and therefore, like many of the mendicant orders, live austerely as they have no income except from alms.  This would help to reaffirm the use of "poor" building materials. The second question refers to the logical use of an earth mortar when covering adobe bricks and a lime mortar when the wall to be covered is made of stone. Therefore, the mortars used are necessarily linked to the support on which they are placed.

Author Response

(The authors gave the same response as above.)

Round 2

Reviewer 1 Report

Comments and Suggestions for Authors

Authors are invited to submit a Response letter where point-by-point all the comments provided are addressed. Moreover, the revised version should highlight in red all the changes reported (including reference section).

Author Response

Dear reviewer

The reviewer is congratulated for the recommendations suggested to improve the comprehension and reading of the research.

The observations made in the paragraphs, tables and graphs both by you and by the other reviewers, have been restructured. The current manuscript goes with the revision format that highlights the changes.

I will have to comment on a few observations that remain.

Point 1.

In the introduction the novelty of the paper should be clearly reported. At least the knowledge process, as unvaluable tool for studying cultural heritage, should be mentioned. Among the others, Author should refer to the following studies, where this topic is fully investigated:

·      Luchin, G., Ramos, L. F., D'Amato, M. (2020). Sonic tomography for masonry walls characterization. International Journal of Architectural Heritage, 14(4), 589-604.

·      Lombillo, I., C. Thomas, L. Villegas, J. P. Fernández-Álvarez, and J. Norambuena-Contreras. 2013. Mechanical characterization of rubble stone masonry walls using non and minor destructive tests, 266–77. Construction and Building Materials. Amsterdam, NL: Elsevier.

Response to Point 1:

Thanks for the references, the research of Luchin et al.2020 as well as Lombillo et al.2013; They are interesting but do not cover the scope of the research.

A mechanical study in the temple of San Francisco is NOT to be carried out, because the church is in operation and is a living museum "they are not ruins" and the religious limited the number of samples that were extracted.

The Sonic Tomography tests could be the most feasible to use in future research, however there were limited resources, so much so that all the tests were carried out outside Ecuador because there are no research centers and the amount of sample extracted was very little.

Point 2.
Introduction should be revised. Some parts me be condensed since information reported are not strictly pertaining to the research work.

Response to Point 2:
The theoretical framework of the research is fed by historical data, an attempt has been made to condense the most important of the construction stages to correlate with the analytical data obtained. Refining historical information, corroborating with the construction stages, and relating them to local materialities is a determinant not visualized in the research.

Point 3.
In the current version the paper just explains the results of some X-rays graphs. More comments should be reported, for instance, considering other similar results already published in literature.

Response to Point 3:
The biggest limitation was the weight of the sample extracted and processed, and with what we had we tested with X-ray diffraction, petrography, and scanning electron microscopy with a microanalysis. It should be noted that the same authors have published works of similar characteristics since 2020, hence a study registered with value in 2003 by Moropoulou.

https://doi.org/10.37558/gec.v17i1.687

https://doi.org/10.3390/min11070781

https://doi.org/10.3390/heritage5040207 

Point 4.
How the results published in this work may be coupled with other non-destructive tests in order to properly identifying the mortar present in the masonry analyzed? This is an important  

Response to Point 4: 

These mineralogical tests give us a first approximation, it is clear that due to limitations they did not allow us to carry out mechanical and physical tests to have a clearer vision. The research made it possible to visit the LNEC in Lisbon and see what would be ambitiously desired to be developed, a compatible replacement mortar and for this, in addition to the characterization and complete analysis (physical, chemical, mechanical and mineralogical), tests must be carried out respecting different protocols, tests and standards.

Point 5.
Conclusion section should be revised reporting only the main outcomes of the work presented.

Response to Point 5: 

Revising, perhaps eliminate the first paragraph of the conclusions because it is not a result by itself of the research, although it provides us with data that validates the historical context of the Viceroyalty of Quito and its constructive history

Reviewer 2 Report

Comments and Suggestions for Authors

After the second review of the paper “Characterization and analysis of the mortars of the church of San Francisco of Quito (Ecuador)”, I have observed that while there have been improvements in the paper's overall quality, especially in terms of readability, there are still some major issues that need to be addressed before it can be considered for publication.

Firstly, I would recommend revising the title to something more precise and academically fitting, such as "Mineralogical and Petrographic Characterization of the Mortars in the Church of San Francisco, Quito, Ecuador."

The introduction section provides an excellent historical background of the monument; however, it fails to establish the problem to be addressed by the study. The authors have not provided any information about why the they want to study these mortars, and what is the aim of the paper and there still no background about the study of the mortars in heritage.

As previously mentioned in my initial review, the historical description should be placed in a separate section following the introduction. In the introduction, the authors have to introduce the research that has been described in the paper (about the importance of the characterization of historic mortars), the problem to solve, and describe previous research in similar cases or other research done in the monument. There is a lot of literature on this topic. If there are no other cases in Ecuador, in other places worldwide…, one of the most important things, which is omitted in the paper, is a clear statement with the aim of the research. A distinct and explicit statement of the research's objectives is crucial and should not be omitted from the paper.

In subsection 2.2, "Methods of Identification," the author mentions the use of a binocular stereo microscope, specifically the Zeiss Stemi 305. However, there is a lack of information regarding the results of observations made using this equipment. What specific aspects of the samples were examined with this microscope, and how did this contribute to the overall analysis? It is not reflected in the paper.

L137 You state that 21 samples were collected but, in the paper, but the analysis only covers data from 11 of these samples. What happened with the others?

In this case, you can obvious this information and indicate that for this research, 11 samples were collected from the site. Or, in any case, you can also justify what were the criteria for taking into consideration only 11 samples instead of 21 and why 10 samples were discarded.

Fig 3. Increase the font size of the text next to the sample ISF-XX, because it is impossible to read. Or, perhaps can fit better in the caption. E.g.: Figure 3. Interior axonometry of the San Francisco church with the locations of the 11 extraction points analysed. ISF-01: South access dome; IDF-02: Niche altarpiece, etc….

L211 Thin slices: The preparation for MOP is better referred to as “thin sections”. Check the whole document.

L238 Perhaps you refer to mineral phases (or mineral species) of the samples because mineral structures refer to the ordination of the atoms in the inner crystallographic configuration.

Table 1. Use the same significant digits in the table (one decimal in this case).

Line 244. …other analysis: specified which ones or avoid this kind of generic statements.

If the mortar type of Table 2 is the results of the MOP observation, XRD data and SEM and EDX analysis, perhaps you can place this table at the end of the section, and indicate this aspect on the caption, because after reading the text it is not clear. In any case, the column Case Study makes no sense. Perhaps you can replace this column with “Construction Phase” to remark that in each phase, the masons used different mortars.

L 267 corrects this: (shown in Figure 6 and Figure 7, respectively)

For better readiness, perhaps you better place the figures just below the text of their description

Figure 6. Thin section image at 200× magnification under POM (with crossed Nichols)

L279 change (b)(c) by (c)(d).

The same must be corrected in Fig. 7, 8 and 9.  

Line 336 kaolinite? It is the first time in the paper that you mentioned the presence of kaolinite in your samples. Where this data come from?

Line 349: Perhaps, you can indicate the mineralogical composition of the material from the quarry (if available) rather than the chemical composition. In this study, you do not have quantified the chemical composition of the samples, so making comparisons may not be advisable as it lacks  absolute geochemical data of both the mortars and the quarries.

Line 351: In what sense contradicts the research of Moropoulou? It is imperative to explain this.

Line 360 What is the meaning of the implication of anorthite for your research? Why did you highlight this finding? Explain a little bit more.

Line 362 Based on both the mineralogical data and the petrographic observations, two types of mortar were identified. One was lime mortar with….

Line 391: tianguez is an Americanism: Explain, in brackets, the meaning in English.

L402 To ensure that the aggregates indeed originate from the quarries of Pintag, further research employing additional techniques such as stable isotopes and fluid inclusion analysis is needed to establish their origin with a higher degree of certainty. While historical documentation suggests that the samples may come from this quarry, this assumption has not been substantiated with analytical data:

The sampling analysis provided an initial insight into the origin of the aggregates, reasonably suggesting that the mortars were supplemented with local raw materials and volcanic-calcitic aggregates. These findings were indicated by the results of the XRD and POM analysis. While these initial findings may suggest a connection to the quarries of Pintag, it is important to note that further, more comprehensive research is required to confirm this link. Historical records do point to Pintag as the source, with references dating back to 1551 when Francisco Ruiz was designated the quarry by the viceroyalty in favor of the Franciscan order. However, scientific evidence is needed to solidify this connection beyond reasonable doubt.

L 411 Amphiboles are not considered clay minerals, but muscovite is already considered in the group of clay minerals.

L411 … The mineralogical nature of the adobe masonry and its coating provide invaluable information not only on the origin and type of the original clay mixture but …. You need to rewrite this sentence since you do not determine the type of clays in the earth mortar (as you indicate in the footnote of table 2: Due to existing limitations in this investigation, it was not possible to perform a clay analysis on the samples ISF-06, ISF-07, ISF-08, and ISF-09).

Author Response

Dear reviewer

The reviewer is congratulated for the recommendations suggested to improve the comprehension and reading of the research.

The observations made in the paragraphs, tables and graphs both by you and by the other reviewers, have been restructured. The current manuscript goes with the revision format that highlights the changes.

I will have to comment on a few observations that remain.

1.- The title will be maintained; it keeps a structure for future research of the historical area of Quito, since the consolidated center for the seventeenth century possessed thirty temples of which so far, we are studying three cases. Although the San Francisco study focuses only on mineralogy and petrography, we hope to continue with more case studies.

2.- The material from the quarry could not be accessed; historically it is known what the quarries were, but they are currently closed, and erosion has removed them. It will have to be stated that the research was originally born only in the main temple, but this grows as the results and analysis and involves more researchers multidisciplinarily, there is a dilemma due to the scope of the doctoral thesis and its original delimitations.

Based on the findings (comparing historical data and new petrographic research from this area) it is possible to corroborate the hypothesis that the Pintag quarry was the deposit that provided material for the temple.

3.- The research of Moropoulou in 2003, focused on another sector of the Franciscan complex that has 3.5 hectares and carried out studies on sections of brick blocks, stone, etc., not on the church cladding by itself.

I know that it is a limitation not to access the original quarry, but at the same time new lines of research are being developed with researchers in the area that will be studied later.

Reviewer 4 Report

Comments and Suggestions for Authors

I'm agree with changes introduced.

Author Response

Dear reviewer

The reviewer is congratulated for the recommendations suggested to improve the comprehension and reading of the research.

The observations made in the paragraphs, tables and graphs both by you and by the other reviewers, have been restructured. The current manuscript goes with the revision format that highlights the changes.

I will have to comment on a few observations that remain

Round 3

Reviewer 1 Report

Comments and Suggestions for Authors

 In the text the revised parts are not highlighted properly.

Author Response

Dear reviewer 1

Thank you for the comments; in this edition all the changes of the document have been recorded both in paragraphs, images, conclusions, and bibliography; as well as the English edited for MDPI service.

Reviewer 2 Report

Comments and Suggestions for Authors

After the third review of the article, and although I believe the authors have barely addressed my recommendations, which undoubtedly would have contributed to an improvement in the article's quality, I consider the current version acceptable for publication.

However, there are minor issues that I think, with little effort on the part of the authors, can be corrected to enhance the article. They have not yet addressed the issue raised in the previous review:

L219. For identification and characterization, several tests were initiated. Thus, with small fragments and in thin sections, the samples were visualized through a binocular stereo microscope Zeiss Stemi 305, and optical microscopy.

I commented in the second review: In subsection 2.2, "Methods of Identification," the author mentions the use of a binocular stereo microscope, specifically the Zeiss Stemi 305. However, there is a lack of information regarding the results of observations made using this equipment. What specific aspects of the samples were examined with this microscope, and how did this contribute to the overall analysis? It is not reflected in the paper.

This info in still missing. I believe it is nonsensical to specify the use of a tool that is not reflected in the research results, as is the case with the binocular microscope, for which its usage is not justified in any section. I suggest correcting this sentence in the methodology section by specifying the characteristics (brand, model, etc.) of the petrographic microscope, not the binocular microscope and remove any reference respect to the use of the binocular microscope.

L317. The authors still fail to explain in what way their findings contradict Moropoulou. While a comment in this regard would be appreciated to enhance the discussion, if they do not intend to provide one, they should consider removing the phrase: 'which contradicts the research by Moropoulou [30].

Regarding the characterization of the original quarries, according to the authors’ comment to my second reviews, perhaps something similar to this sentence could be added to the conclusion section:

Based on the findings (comparing historical data and new petrographic research from this area) it is quite possible to corroborate the hypothesis that the Pintag quarry was the deposit that provided material for the temple. Although the material from the original quarries is currently not accessible, historically it is known which quarries were, but they are currently closed, and erosion has eliminated them. However, as the research progresses, an attempt could be made to search for original material to study its evolution over time, and confirm the origin of the mortars used in this case study.

L 138 carried out; However: use period or remove caps in however.

Author Response

Dear reviewer 2

Thank you for the comments; in this edition all the changes of the document have been recorded both in paragraphs, images, conclusions, and bibliography; as well as the English edited for MDPI service.

The authors have corrected the paragraph of Antonia Moropoulou that was noted by the reviewer.

The authors have restructured the arguments of the binocular stereo microscope in the sections related to it with the aim of being clearer in the contribution of research inputs.

The conclusion has been restructured with respect to the analysis of possible historical quarries for the extraction of the material.

Errors in lines L317 and L338 have been corrected (note that the numbering has been changed in this edition of the document)